# Analyzing Classification Performance of fNIRS-BCI for Gait Rehabilitation Using Deep Neural Networks

**DOI:** 10.3390/s22051932

**Published:** 2022-03-01

**Authors:** Huma Hamid, Noman Naseer, Hammad Nazeer, Muhammad Jawad Khan, Rayyan Azam Khan, Umar Shahbaz Khan

**Affiliations:** 1Department of Mechatronics and Biomedical Engineering, Air University, Islamabad 44000, Pakistan; humahamid244@gmail.com (H.H.); hammad@mail.au.edu.pk (H.N.); 2School of Mechanical and Manufacturing Engineering, National University of Science and Technology, Islamabad 44000, Pakistan; jawad.khan@smme.nust.edu.pk; 3Department of Mechanical Engineering, University of Saskatchewan, Saskatoon, SK S7N 5A9, Canada; rayyan.khan@usask.ca; 4Department of Mechatronics Engineering, National University of Sciences and Technology, Islamabad 44000, Pakistan; u.shahbaz@ceme.nust.edu.pk; 5National Centre of Robotics and Automation (NCRA), Rawalpindi 46000, Pakistan

**Keywords:** functional near-infrared spectroscopy, brain-computer interface, convolutional neural network, long short-term memory, neurorehabilitation

## Abstract

This research presents a brain-computer interface (BCI) framework for brain signal classification using deep learning (DL) and machine learning (ML) approaches on functional near-infrared spectroscopy (fNIRS) signals. fNIRS signals of motor execution for walking and rest tasks are acquired from the primary motor cortex in the brain’s left hemisphere for nine subjects. DL algorithms, including convolutional neural networks (CNNs), long short-term memory (LSTM), and bidirectional LSTM (Bi-LSTM) are used to achieve average classification accuracies of 88.50%, 84.24%, and 85.13%, respectively. For comparison purposes, three conventional ML algorithms, support vector machine (SVM), k-nearest neighbor (k-NN), and linear discriminant analysis (LDA) are also used for classification, resulting in average classification accuracies of 73.91%, 74.24%, and 65.85%, respectively. This study successfully demonstrates that the enhanced performance of fNIRS-BCI can be achieved in terms of classification accuracy using DL approaches compared to conventional ML approaches. Furthermore, the control commands generated by these classifiers can be used to initiate and stop the gait cycle of the lower limb exoskeleton for gait rehabilitation.

## 1. Introduction

The world has been striving to create a communication channel based on signals obtained from the brain. A brain-computer interface (BCI) is a communication system that provides its users with control channels independent of the brain’s output channel to control external devices using brain activity [1,2]. The BCI system was first introduced by Vidal in 1973 in which he proposed three assumptions regarding BCI, including analysis of complex data in the form of small wavelets [3]. A typical BCI system consists of five stages, as shown in Figure 1. The first stage is the brain-signal acquisition using a neuroimaging modality. The second is preprocessing those signals as they contain physiological noises and motion artefacts [4]. The third stage is feature extraction in which meaningful features are selected [5]. These features are then classified using suitable classifiers. The final stage is the application interface in which the classified BCI signals are given to an external device as a control command [6].

Researchers have been using different techniques to acquire brain signals [7]. These techniques include electroencephalography (EEG), functional near-infrared spectroscopy (fNIRS), magnetoencephalography (MEG), and functional magnetic resonance imaging (fMRI) [8]. EEG is a technique used to analyze brain activity by measuring changes in the electrical activity of the active neurons in the brain [9], while MEG measures the changes in magnetic fields associated with the brain’s electrical activity changes [10]. fMRI is another modality for analyzing brain function by measuring the localized changes in cerebral blood flow stimulated by cognitive, sensory, or motor tasks [11,12]. In this study, we will only be dealing with fNIRS-BCI. fNIRS is a non-invasive optical neuroimaging technique that measures the concentration changes of oxy-hemoglobin (ΔHbO) and deoxy-hemoglobin (ΔHbR) that are associated with the brain activity stimulated, when participants perform experiments, such as arithmetic tasks, motor imagery, motor execution, etc. [13]. It is a non-invasive, portable and easy-to-use brain imaging technique that helps study the brain’s functions in the laboratory, naturalistic, and real-world settings [14]. fNIRS consists of near-infrared light emitter detector pairs. The emitter emits light with several distinct wavelengths absorbed in the subject’s scalp, consequently causing scattered photons; while some of them are absorbed, the others disperse and pass through the cortical areas where HbO and HbR chromophores absorb the light and have different absorption coefficients. The concentration of HbO and HbR changes along the photon path in consideration of the modified Beer-Lambert law [15].

In the recent decade, the research on fNIRS-BCI has increased enormously, and new diverse techniques, particularly in its applications, may grow exponentially over the following years [16]. One of the significant fields of fNIRS application is cognitive neuroscience, particularly in real-world cognition [17], neuroergonomics [18], neurodevelopment [19], neurorehabilitation [20], and in social interactions. fNIRS-BCI can provide a modest input for BCI systems in the real time, but improvements are required for this system as there is the difficulty faced with the collection and interpretation of the data for classifiers due to noise in the data and subject variations [21].

Well-designed wearable assistive devices for rehabilitation and performance augmentation purposes have been developed that run independently of physical or muscular interventions [22,23,24]. Recent studies focus on acquiring the user’s intent through brain signals to control these devices/limbs [25,26,27]. In assistive technologies, the fNIRS-BCI system is a suitable technique for controlling exoskeletons and wearable robots by using intuitive brain signals instead of being controlled manually by various buttons in order to get the desired postures [28]. In addition, it has a better spatial resolution, fewer artefacts, and acceptable temporal resolution, which makes it a suitable choice for rehabilitation and mental task applications [29].

To find the best machine-learning (ML) method for fNIRS-based active-walking classification, a series of experiments with various ML algorithms and configurations were conducted; the classification accuracy achieved was above 95% [30] for classifying 1000 samples using different ML algorithms, such as random forest, decision tree, logistic regression, support vector machine (SVM) and k-nearest neighbor (k-NN). In order to achieve minimum execution delay and minimum computation cost for an online BCI system, linear discriminant analysis (LDA) was used with combinations of six features for walking intention and rest tasks [31].

The traditional method of extracting and selecting acceptable features can result in performance degradation. In contrast, deep neural networks (DNNs) can extract different features from raw fNIRS signals, nullifying the need for a manual feature extraction stage in the BCI system development, but limited studies are available so far [32,33].

Convolutional neural network (CNN) is a deep-learning (DL) approach that can automatically learn relevant features from the input data [34]. In a study, CNN architecture was compared to conventional ML algorithms, and CNNs performed better in terms of classification with an average classification accuracy of 72.35 ± 4.4% for the four-class motor imagery fNIRS signals [35]. CNN-based time series classification (TSC) methods to classify fNIRS-BCI are compared with ML methods, such as SVM. The results showed that the CNN-based methods performed better in terms of classification accuracy for left-handed and right-handed motor imagery tasks and achieved up to 98.6% accuracy [36].

Time-series data can be handled more precisely using long short-term memory (LSTMs) modules. In a study, four DL models were evaluated including multilayer perceptron (MLP), forward and backward long short-term memory (LSTMs), and bidirectional LSTM (Bi-LSTM) for the assessment of human pain in nonverbal patients, and Bi-LSTM model achieved the highest classification accuracy of 90.6% [37]. Using the LSTM network, large time scale connectivity can be determined with the help of the InceptionTime neural network, which is an attention LSTM neural network utilized for the brain activations of mood disorders [38]. A recent study assessed four-level mental workload states using DNNs, such as CNNs and LSTM for fNIRS-BCI system, with average classification accuracies of 87.45% and 89.31% [39].

This study contributes to the development of a neurorehablitation tool in the gait training of elderly and disabled people. The main objective of this study is to compare two classification approaches, ML and DL, to achieve high performance in terms of classification accuracy on the time-series fNIRS data. The complete summary of the methods employed in this research is depicted in Figure 2.

## 2. Materials and Methods

### 2.1. Experimental Paradigm

Nine healthy right-handed male subjects of 27 ± 5 median age were selected. The subjects had no history of motor disability or any visual neurological disorders affecting the experimental results. fNIRS-based BCI signals were acquired from the primary motor cortex (M1) in the left hemisphere for self-paced walking [40]. Before the start of each experiment, the subjects were asked to take a rest for 30 s in a quiet room to establish the baseline condition; it was followed by 10 s of walking on a treadmill, followed by 20 s of rest while standing on the treadmill. At the end of each experiment, a 30 s rest was also given for baseline correction of the signals. Each subject performed 10 trials, as shown in Figure 3. Excluding the initial (30 s) and final (30 s) of rest, the total length of each experiment was 300 s for each subject. All the experiments were conducted in accordance with the latest Declaration of Helsinki and approved by the Institutional Review Board of Pusan National University, Busan, Republic of Korea [41].

### 2.2. Experimental Configuration

In this study a multi-channel continuous-wave imaging system (DYNOT: Dynamic Near-infrared Optical Tomography; NIRx Medical Technologies, NY, USA) was used to acquire the brain signals, which operate on two wavelengths, 760 and 830 nm, with a 1.81 Hz sampling frequency. Four near-infrared light detectors and five sources (total of nine optodes) were placed on the left hemisphere of the M1 with 3 cm of distance between a source and a detector [42]. A total of twelve channels were formed in accordance with the defined configuration, as shown in Figure 4.

### 2.3. Signal Acquisition

The acquired light intensity signals from the left hemisphere of the M1 were first converted into oxy- and deoxy-hemoglobin concentration changes ($\Delta c_{HbO}\left(t\right)$, and $\Delta c_{HbR}\left(t\right)$) using the modified Beer-Lambert law [43].
(1)$$\left[\frac{\Delta c_{HbO}\left(t\right)}{\Delta c_{HbR}\left(t\right)}\right]=\frac{{\left[\begin{array}{cc} \alpha_{HbO}\left(\lambda_{1}\right) & \alpha_{HbR}\left(\lambda_{1}\right)\\ \alpha_{HbO}\left(\lambda_{2}\right) & \alpha_{HbR}\left(\lambda_{2}\right) \end{array}\right]}^{-1}\left[\begin{array}{l} \Delta A\left(t,\lambda_{1}\right)\\ \Delta A\left(t,\lambda_{2}\right) \end{array}\right]}{d*l}$$
where $\Delta c_{HbO}\left(t\right)$ and $\Delta c_{HbR}\left(t\right)$ are the concentration changes of HbO and HbR in [μM], *A(t,*
$\lambda_{1}$*)* and *A(t,*
$\lambda_{2}$*)* are the absorptions at two different instants, l is the emitter–detector distance (in millimeters), *d* is the unitless differential path length factor (DPF), and $\alpha_{HbO}\left(\;\lambda \right)$ and $\alpha_{HbR}\left(\lambda \right)$ are the extinction coefficients of HbO and HbR in [μM^−1^ cm^−1^].

### 2.4. Signal Processing

After obtaining oxy-hemoglobin concentration changes ($\Delta c_{HbO}\left(t\right)$ and $\Delta c_{HbR}\left(t\right)$), the brain signals acquired were filtered with suitable filters using the modified Beer-Lambert law. In order to minimize the physiological or instrumental noises, such as heartbeat (1–1.5 Hz), respiration (~0.5 Hz), Mayer waves (blood pressure), artefacts, and others, the signals were low-pass filtered at a cut-off frequency of 0.5 Hz and a high-pass filter with cut-off frequency of 0.01 Hz [44]. The filter used for $\Delta c_{HbO}\left(t\right)$ signals were hemodynamic response (hrf) using NIRS-SPM toolbox [45]. The averaged $\Delta c_{HbO}\left(t\right)$ signal for task and rest of subject 1 after filtering is shown in Figure 5.

### 2.5. Feature Extraction

For the conventional ML algorithms, five different features of filtered $\Delta c_{HbO}\left(t\right)$ signals were extracted using the spatial average for all 12 channels. Five statistical properties (mean, variance, skewness, kurtosis, and peak) of the averaged signals were calculated for the entire task and rest sessions. In this study, a feature combination of signal mean, signal peak, and signal variance a was used for the ML classifiers. This specific combination was selected based on the higher classification accuracies that were obtained using these features [46,47].

For the mean, the equation was as follows:(2)$$\bar{X}=\frac{1}{N}\sum_{i=1}^{N}\left(Z_{i}\right)$$
where *N* is the total number of observations, and $Z_{i}$ is the $\Delta c_{HbO}\left(t\right)$ across each observation. For signal variance, the calculation was as follows:(3)$$\sigma^{2}=\frac{\sum_{i=1}^{N}\left(X_{i}-\bar{X}\right)}{n-1}$$
where *n* is the sample size, $X_{i}$ is the *i*th element of the sample, and $\bar{X}$ is the mean of the sample. To calculate signal peak, the max function in MATLAB^®^ was used.

## 3. Classification Using Machine-Learning Algorithms

### 3.1. Support Vector Machine (SVM)

SVM is a commonly used classification technique suitable for fNIRS-BCI systems for handling high-dimensional data [48,49]. In supervised learning, the SVM classifier creates hyperplanes to maximize the distance between the separating hyperplanes and the closest training points [50]. The hyperplanes, known as the class vectors, are called support vectors. The separating hyperplane in the two-dimension features space is given by:(4)$$f\;\left(x\right)=r\;\cdot \;x+b$$
where *b* is a scaling factor, and $r,\;x\in R^{2}\;\mathrm{and}\;b\in R^{1}$. The optimal solution, *r**, that is the distance between the hyperplane and the nearest training point(s) is maximized by minimizing the cost function. The optimal solution, *r** is given by the equation.

Minimize
(5)$$\frac{1}{2}\;{\left|\left|w\right|\right|}^{2}+C\;\sum_{i=1}^{n}\xi_{i}\;$$

Subject to
(6)$$y_{i}\left(\;w^{T}\;x_{i}+b\right)\;\geq \;1\;–\;\xi_{i},\;\;\;\;\;\;\xi_{i}\;\geq \;0$$
where $y_{i}$ represents the class label for the *i*th sample, *T* is the transpose, and *n* is the total number of samples, ${\left|\left|w\right|\right|}^{2}\;$= $w^{T}w$. where $w^{T}$ and $x_{i}$ $\in R^{2}$, $b\in R^{1}$, *C* is the trade-off parameter between the margin and error, and $\xi_{i}$ is the training error.

### 3.2. k-Nearest Neighbor (k-NN)

k-NN predicts the test sample’s category; the k value represents the number of neighbors considered and classifies it in the same class as its nearest neighbor based upon the largest category probability [51]. Euclidean distance is the distance between the trained and the test object given by the equation.
(7)$$D_{E}\;\left(p,q\right)=\sqrt{\sum_{i=1}^{n}{\left(p_{i}-q_{i}\right)}^{2}}$$
where *n* is the *n*-space, p and q are two points in the Euclidean *n*-space, and $p_{i},q_{i}$ are the two vectors, stating from the origin of the space.

k-NN is a widely used efficient classification method for BCI applications because of its low computational requirements and simple implementation [52,53].

### 3.3. Linear Discriminant Analysis (LDA)

LDA has discriminant hyperplanes to separate classes from each other. LDA performs well in various BCI systems because of its simplicity and execution speed [54]. It minimizes the variance of the class and maximizes the separation between the mean of the class by maximizing the Fisher’s criterion [55]. The equation for Fisher’s criterion is given by:(8)$$J\;\left(v\right)=\frac{v^{T}\;S_{b}v}{v^{T}\;S_{w}v}$$
where $S_{b}$ and $S_{w}$ are the between-class and within-class scatter matrices given as:(9)$$S_{b}=\left(m_{1}-m_{2}\right){\left(m_{1}-m_{2}\right)}^{T},\;$$
$$S_{w}=\underset{x_{n}\in \;C_{1}}{\sum }\left(x_{n}-m_{1}\right){\left(x-m_{2}\right)}^{T}+\underset{x_{n}\in \;C_{2}}{\sum }\left(x_{n}-m_{1}\right){\left(x-m_{2}\right)}^{T}$$
where $x_{n}$ denotes samples, $m_{1}$ and $m_{2}$ are the group means of classes $C_{1}$ and $C_{2}$, respectively.

## 4. Classification Using Deep-Learning Algorithms

fNIRS signal classification with conventional ML methods is composed of local and global feature extraction, e.g., independent component analysis (ICA) and principal component analysis (PCA), selection of possible features, their combinations, and dimensionality reduction, which leads to the biasness and overfitting of the data [56]. It is because of these limitations a large amount of time is consumed in the mining and preprocessing of the data [57].

The BCI classification task can avoid local filtering, noise removal, and manual local feature extraction by utilizing DL algorithms as an alternative to avoid the need for manual feature engineering, data cleaning, analysis, transformation, and dimensionality reduction before feeding it to the learning machines [58]. Extracting and selecting appropriate features is critical with fNIRS-BCI signals, and the multi-dimensionality and complexity of fNIRS signals make it ideal for DL algorithms to work with.

### 4.1. Convolutional Neural Networks (CNNs)

CNNs are a type of neural networks that are capable of automatically learning appropriate features from the input fNIRS time-series data. CNNs consist of several layers, such as the convolutional layer, pooling layer, fully connected layer, and output layer [59]. The input fNIRS time-series data (the changes in the HbO concentrations) across all the channels are passed through CNN layers. The convolutional layer contains filters that are known as convolution kernels to extract features. CNN minimizes the classification errors by adjusting the weight parameters of each filter using forward and backward propagation.

The convolution operation is the sliding of a filter over the time series, which results in activation maps also known as feature maps that stores the features and patterns of the fNIRS data [60]. Convolution operation for time stamp *t* is given by the equation:(10)$$\;\;\;C_{t}=f\;\left(\omega \;*\;X_{t-l/2:t+l/2}+b\right)\;|\;\forall \;t\;\in \;\left[1,\;T\;\right]$$
where *C* is the output of a convolution (dot product) on a time series, *X*, of length, *T*, with a filter, ω, of length, *l*, *b* is a bias parameter, and *f* is a non-linear function, such as the rectified linear unit (ReLU).

After the convolutional layer, we have a pooling layer that downsamples the spatial size of the data and also reduces the number of parameters in the network [61]. The pooling layer is followed by a fully connected layer, as shown in Figure 6 in which each data point is treated as a single neuron that outputs the class scores, and each neuron is connected to all the neurons in the previous layer [62].

The proposed CNN model consists of the input layer, three one-dimensional convolutional layers, max-pooling layers, dropout layers, a fully connected layer, and an output layer. The three convolutional layers contain filters 128, 64, 32 with a kernel size of 3, 5, 11, respectively. A dropout layer of 0.5 ‘dropout ratio’ was added after each convolutional layer to avoid overfitting, followed by a pooling layer with a stride of two. The input fNIRS time-series data after passing through a number of convolutional, max-pooling, and dropout layers is flattened and fed into the fully connected layers for the purpose of classification. The fully connected layer of 100 units is incorporated with ReLU activation. The output layer consists of two neurons corresponding to the two classes with Softmax activation. The optimization technique used was Adam with a batch size of 150 and 500 number of epochs.

### 4.2. Long Short-Term Memory (LSTM) and Bi-LSTM

LSTM is a DL algorithm that can achieve high accuracies in terms of classification, processing, and forecasting predictions on the time-series fNIRS data. LSTMs have internal mechanisms called gates that can regulate the flow of information [63]. These gates, such as forget gate, input gate, and output gate, can learn which data in a sequence are necessary to keep or throw away [64]. By doing that, it can pass relevant information down the long chain of sequences to make predictions. The equations for forget gate ($f_{t}$), input gate ($i_{t}$) and output gate ($o_{t}$) are given by:(11)$$f_{t}=\;\sigma \left(W_{f}\;\cdot \;\left[h_{t-1},x_{t}\;\right]+b_{f}\right)$$
(12)$$i_{t}=\;\sigma \left(W_{i}\;\cdot \;\left[h_{t-1},x_{t}\;\right]+b_{i}\right)$$
(13)$$o_{t}=\;\sigma \left(W_{o}\;\cdot \;\left[h_{t-1},x_{t}\;\right]+b_{o}\right)$$
where $W_{f}$, $W_{i}$, and $W_{o}$ are the weight matrices of forget, input, and output gates, respectively, and $h_{t-1}$ is the hidden state.

These gates contain sigmoid and Tanh activations to help regulate the values flowing through the network [65]. General sigmoid function is given by:(14)$$f\left(x\right)=\frac{1}{1+e^{-k\left(x-x_{o}\right)}}$$
where $x_{o}$ is the *x*-value of the sigmoid midpoint, *e* is the natural logarithm base, and *k* is the growth rate.

For LSTM the data has to be reshaped because it expects the data in the form of (*m* × *k* × *n*), where *m* is the number of samples, *n* refers to the number of fNIRS channels (12 $\Delta c_{HbO}\left(t\right)$ channels), and *k* refers to the time steps. The proposed LSTM model consisted of an input layer, four LSTM layers, a fully connected layer, and an output layer, as shown in Figure 7. A dropout layer of 0.5 ‘dropout ratio’ was added after the LSTM layers to avoid overfitting. The output from the dropout layer is flattened and fed to the fully connected layer of 64 units, also known as the dense layer, and incorporated with ReLU activation. Finally, an output layer consists of two neurons corresponding to the two classes with Softmax activation. The early-stopping technique was used to avoid overfitting with the patience of 50; a batch size of 150 over 500 epochs with Adam optimization technique.

Bi-LSTM models are a combination of both forward and backward LSTMs [66]. These models can run inputs in two ways, from past to future and from future to past and have both forward and backward information about the sequence at every time step [67]. Bi-LSTM differs from conventional LSTMs as they are unidirectional, and with bidirectional, we are able at any point in time to preserve information from both past and future, which is why they perform better than conventional LSTMs [68].

The proposed Bi-LSTM model consisted of two Bi-LSTM layers with 64 hidden units, a fully connected layer, and an output layer, as shown in Figure 8. The fully connected layer of 64 units is employed with ReLU activation, and the output layer consists of two neurons corresponding to the two classes with Softmax activation.

## 5. Results

The results evaluated for all the methods used in this study are presented in this section, including the validation of the methods. ML algorithms (SVM, k-NN, and LDA) were performed on MATLAB^®^ 2020a Classification Learner App, whereas DL algorithms (CNN, LSTM, and Bi-LSTM) were performed on Python 3.7.12 on Google Colab using Keras models with TensorFlow.

### 5.1. Classification Accuracy of Machine-Learning Algorithms

For ML algorithms, five features (signal mean, signal variance, signal skewness, signal kurtosis, and signal peak) across all 12 channels of filtered $\Delta c_{HbO}\left(t\right)$ signals were spatially calculated. Three feature combinations that were signal mean, signal variance, and signal peak yielded the best results. The manually extracted features from fNIRS data of walking and rest states of nine subjects are fed to the three conventional ML algorithms, SVM, k-NN, and linear LDA, and the highest accuracies obtained were 78.90%, 77.01%, and 66.70% across 12 channels, respectively, as given in Table 1.

To curb overfitting, 10-fold cross-validation was used for the training of the models. The accuracy of conventional ML algorithms for all nine subjects is shown in a bar graph in Figure 9.

### 5.2. Classification Accuracy of Deep Learning Algorithms

To evaluate the deep-learning models, the dataset was initially split into an 80:20 ratio, the training set (80%) and the testing set (20%). The training set used for DL methods in this study has 12 feature dimensions. The learning performance of the algorithm is affected by the size of the training set, which is why 20% of the validation set were employed for each subject since the larger training set usually provides higher classification performance [69]. Although, for CNN, a smaller number of samples after the 30%validation set also attained classification accuracy reaching 90%. The pre-processed fNIRS data of walking and rest states of nine subjects is fed to the three DL algorithms, CNN, LSTM, and Bi-LSTM; the highest classification accuracies obtained were 95.47%, 95.35%, and 95.54% across 12 channels, respectively. The classification accuracy of DL algorithms for all nine subjects is shown in a bar graph in Figure 10.

All the DL models (CNN, LSTM, and Bi-LSTM) were compiled with the metric “accuracy”, which is the measure of the number of correct predictions from all the predictions that were made. To further evaluate the effectiveness of the model, model “precision”, which is the number of positive predictions divided by the total number of positive predicted values and model “recall”, which is the number of actual positives divided by the total number of positive values were also calculated. Accuracy, precision, and recall of DL algorithms are summarized in Table 2, Table 3 and Table 4. The loss function used for the models was “categorical cross-entropy” which is a measure of prediction error, and the optimization technique used was “Adam optimizer”. In order to avoid overfitting, early-stopping technique was used to halt the training when the error during the last 50 epochs is not reduced. Learning rate of 0.001 and decay factor of 0.5 were used in all DL models.

To evaluate the statistical significance of ML and DL methods, a *t*-test was performed for the best performing DL method (CNN) and all the other five classifiers accuracies, as shown in Table 5 The results of these statistical tests showed significant improvement of classification accuracy for the proposed DL method (*p* < 0.008) and the null hypothesis, meaning no statistical significance was rejected.

### 5.3. Validation

For the purpose of validation of the proposed methods, the analysis was also performed on an open-access database containing fNIRS brain signals ($\Delta c_{HbO}\left(t\right)$ and $\Delta c_{HbR}\left(t\right)$) for the dominant foot tapping vs. rest [70]. The analysis was performed for 20 subjects with 25 trials for each subject. By applying the ML methods (SVM, k-NN, and LDA) on the fNIRS dataset for the dominant foot tapping vs. rest tasks, the average classification accuracies were estimated at 66.63%, 68.38%, and 65.96%, respectively, while for DL methods (CNN, LSTM, and Bi-LSTM) the average classification accuracies were estimated at 79.726%, 77.21%, and 78.97%, respectively. The students’ *t*-test showed significantly higher performance (*p* < 0.008) for the proposed DL method. The results obtained from the validation dataset also confirmed the high performance of the proposed DL methods over the conventional ML methods.

## 6. Discussion

Around the world, there are a substantial number of people that have gait impairment and permanent disability in their lower limbs [71]. In the recent decade, the development of wearable/portable assistive devices for mobility rehabilitation and performance augmentation focuses on acquiring the user’s intent through brain signals to control these devices/limbs [72]. In the field of assistive technologies, the fNIRS-BCI system is a relatively suitable technique for the control of exoskeletons and wearable robots by using intuitive brain signals instead of being controlled manually by various buttons to get the desired postures [28,31]. It has a better spatial resolution, fewer artefacts, and acceptable temporal resolution, which makes it a suitable choice for rehabilitation and mental task applications [29,73]. High accuracy BCI systems are to be designed in order to improve the quality of life of people with gait impairment since any misclassification can probably result in a serious accident [56]. To achieve this, the proposed DL and conventional ML methods are investigated for a state-of-the-art fNIRS-BCI system. The control commands generated through these models can be used to initiate and stop the gait cycle of the lower limb exoskeleton for gait rehabilitation.

Researchers have been exploring different ways to improve the classification accuracies by using different feature extraction techniques, feature combinations, or by using different machine-learning algorithms [30]. In a study, six feature combinations, signal mean (SM), signal slope (SS), signal variance (SV), slope kurtosis (KR), and signal peak (SP) have been used for walking and rest data, and the highest average classification accuracy of 75% was obtained from SVM using the hrf filter [31]. In this study, we used three feature combinations of the signal mean, signal variance, and signal peak, and the accuracy obtained from SVM using these features were 73.91%. In a recent study, four-level, mental workload states were assessed using the fNIRS-BCI system by utilizing DNNs, such as CNN and LSTM, and the average accuracy obtained using CNN was 87.45% [39]. Our study achieved almost the same average classification accuracy for CNN with 87.06% for two-class motor execution of walking and rest tasks.

CNN generally refers to a two-dimensional CNN used for image classification, in which the kernel slides along two dimensions on the image data. Recently, researchers have started using deep learning for fNIRS-BCI and bioinformatics problems and have achieved good performances using 2-D CNNs [35,74]. However, in this study, we have considered the one-dimensional CNN for time series fNIRS signals of motor execution tasks and reached a satisfactory classification accuracy. The highest average classification accuracy obtained in this study is 88.50%. For time-series fNIRS data, LSTMs and Bi-LSTMs can achieve high accuracy in terms of classification, processing, and forecasting predictions. In a study for assessing human pain in nonverbal patients, LSTM and Bi-LSTM models were evaluated, and the Bi-LSTM model achieved the highest classification accuracy of 90.6% [37]. In another study, the LSTM based conditional generative adversarial network (CGAN) system was analyzed to determine whether the subject’s task is left-hand finger tapping, right-hand finger tapping, or foot tapping based on the fNIRS data patterns, and the classification accuracy obtained was 90.2%. In the present study, the highest accuracy achieved on any subject with LSTM and Bi-LSTM is 95.35% and 95.54%, respectively, across all 12 channels. The comparison of the average accuracies obtained using ML and DL approaches is shown in a bar graph in Figure 11.

## 7. Conclusions

In this study, two approaches, ML and DL, are investigated to decode two-class data of walking and rest tasks to obtain maximum classification accuracy. The DL approaches proposed in this study, CNN, LSTM, and Bi-LSTM, attained enhanced performance of the fNIRS-BCI system in terms of classification accuracy as compared to conventional ML algorithms across all nine subjects. The highest average classification accuracy of 88.50% was obtained using CNN. CNN showed significantly (p<0.008) better performance as compared to all other ML and DL algorithms. The control commands generated by the classifiers can be used to start and stop the gait cycle of the lower limb exoskeleton which can effectively assist elderly and disabled people in the gait training.

## Figures and Tables

**Figure 1 sensors-22-01932-f001:**
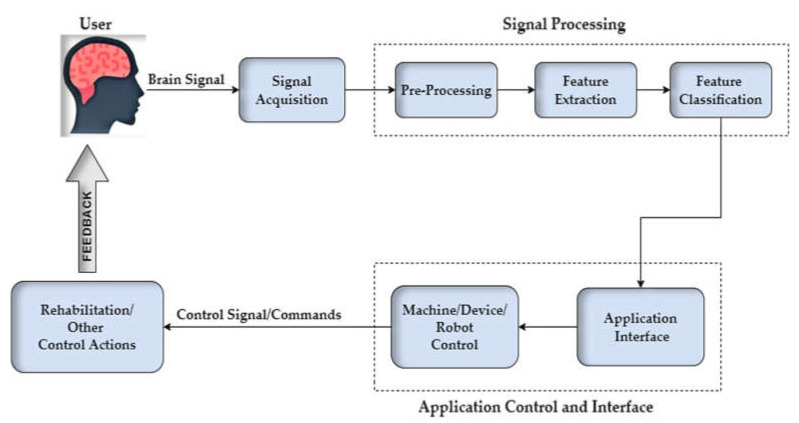
Basic design and operation of the brain-computer interface (BCI)-based control.

**Figure 2 sensors-22-01932-f002:**
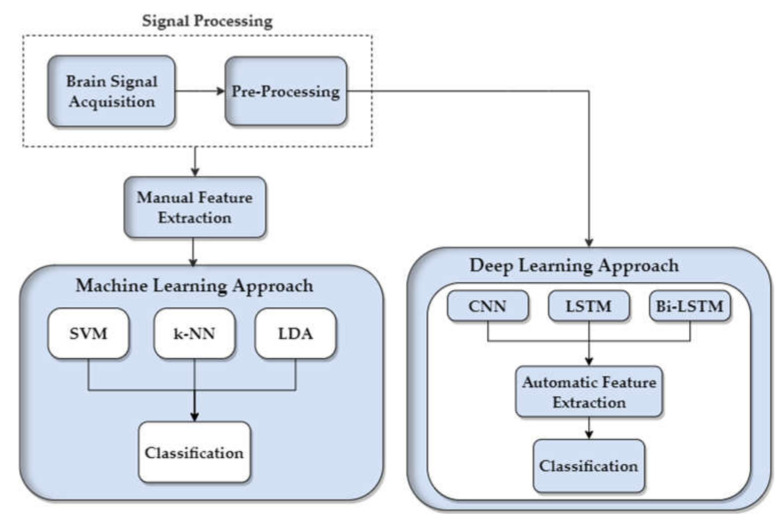
Time-series functional near-infrared spectroscopy (fNIRS) signal classification for walking and rest tasks using conventional machine-learning (ML) and DL algorithms. Signal processing and feature engineering followed by classification using ML and DL approaches.

**Figure 3 sensors-22-01932-f003:**
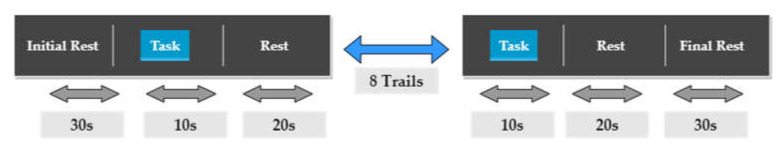
Experimental paradigm with experimental 10 trials with initial and final 30 s rest.

**Figure 4 sensors-22-01932-f004:**
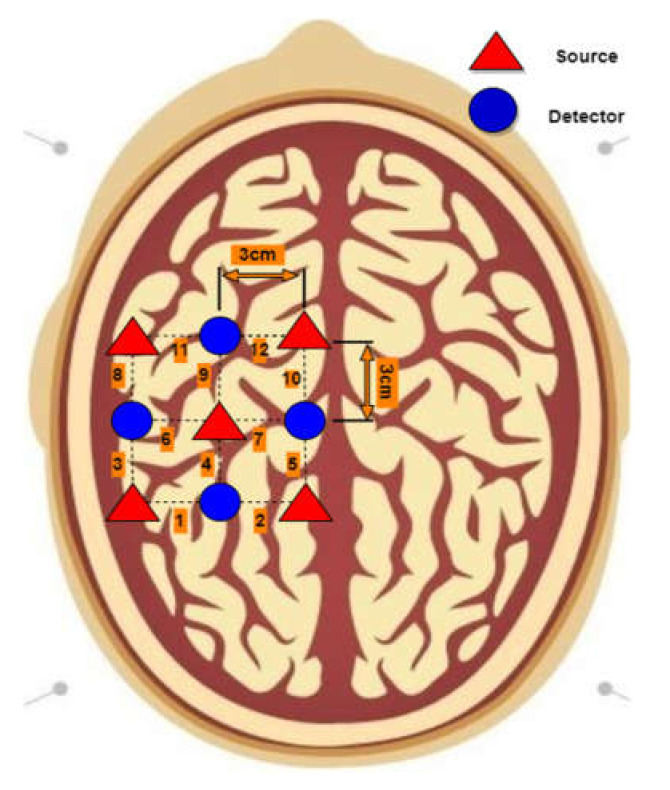
Optode placement on the left hemisphere of the motor cortex with channel configuration using 10–20 international system.

**Figure 5 sensors-22-01932-f005:**
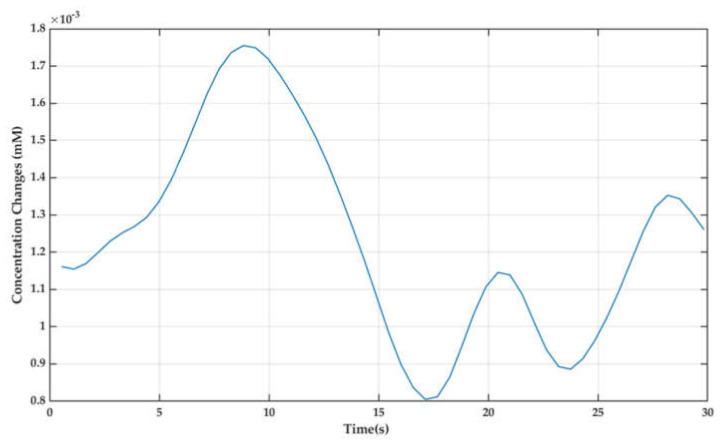
Averaged $\Delta c_{HbO}\left(t\right)$ signal for task and rest of subject 1.

**Figure 6 sensors-22-01932-f006:**
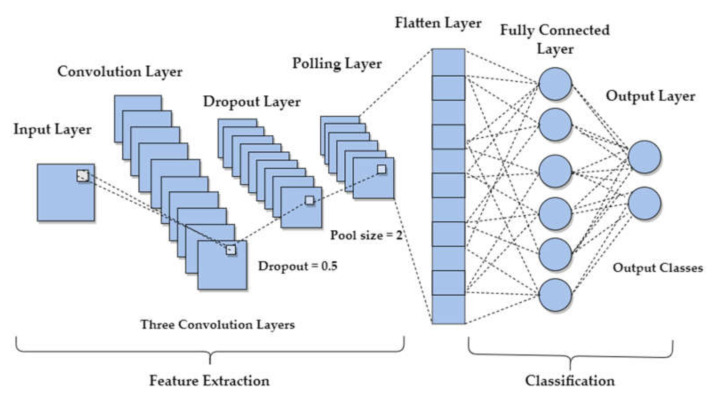
Convolutional neural network (CNN) model with convolutional layer, dropout layer, pooling layer, flatten layer, fully connected layer, and output layer.

**Figure 7 sensors-22-01932-f007:**
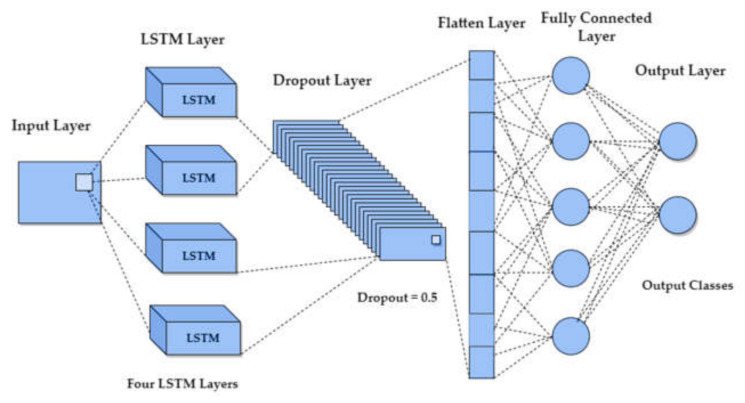
Long short-term memory (LSTM) model with four LSTM layers, dropout layer, flatten layer, fully connected layer, and output layer.

**Figure 8 sensors-22-01932-f008:**
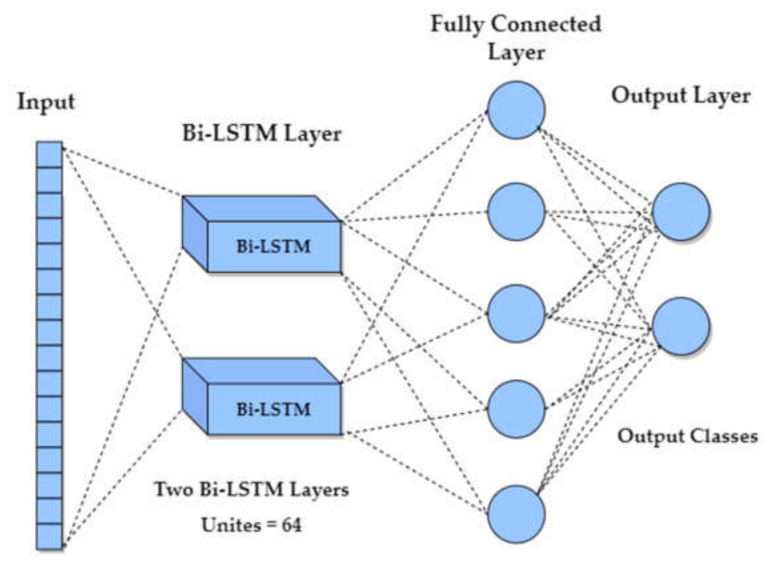
Bidirectional LSTM (Bi-LSTM) model with two Bi-LSTM layers with 64 units, fully connected layer, and output layer.

**Figure 9 sensors-22-01932-f009:**
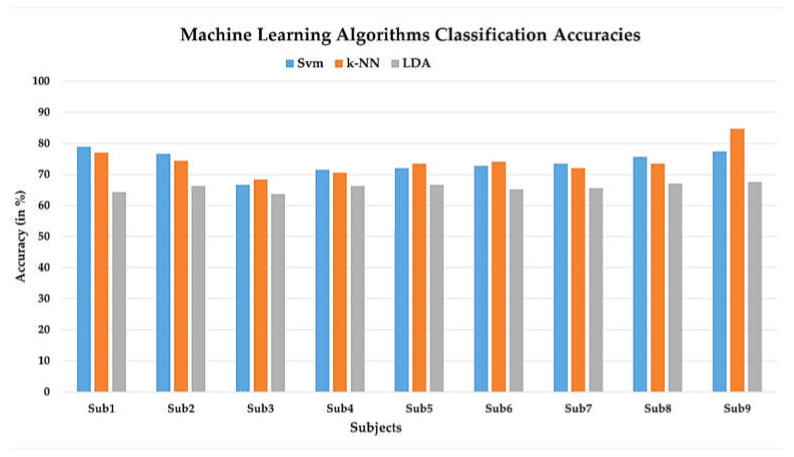
Machine-learning (ML) classifier accuracies (in %) for nine subjects. The ML classifiers are support vector machine (SVM), k-nearest neighbor (k-NN), and linear discriminant analysis (LDA).

**Figure 10 sensors-22-01932-f010:**
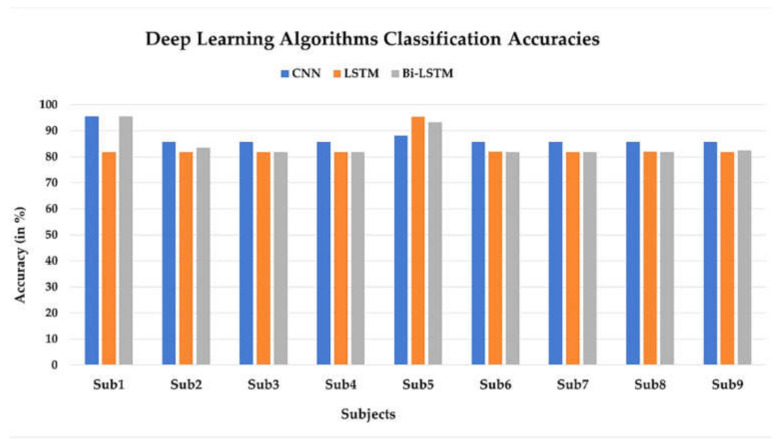
Deep-learning (DL) classifier accuracies (in %) for nine subjects. The DL classifiers are convolutional neural network (CNN), long short-term memory (LSTM), and bidirectional LSTM (Bi-LSTM).

**Figure 11 sensors-22-01932-f011:**
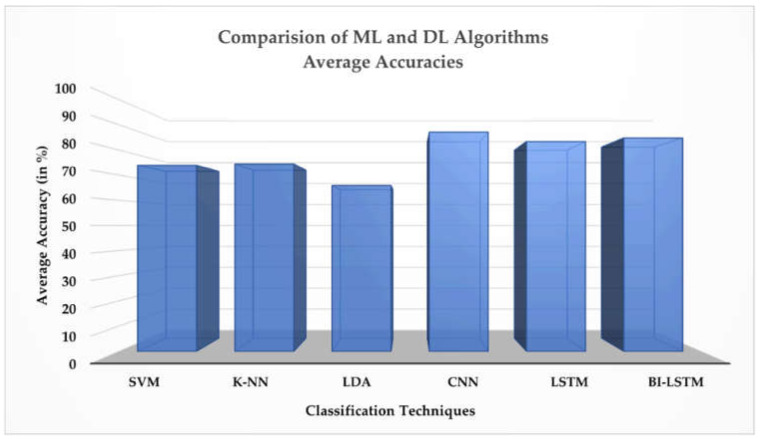
Comparison between machine-learning (ML) classifiers (support vector machine (SVM), k-nearest neighbor (k-NN), and linear discriminant analysis (LDA)) and deep-learning (DL) classifiers (convolutional neural network (CNN), long short-term memory (LSTM), and bidirectional LSTM (Bi-LSTM)) based on average accuracies.

**Table 1 sensors-22-01932-t001:** Accuracy of conventional machine-learning (ML) algorithms, k-nearest neighbor (k-NN), support vector machine (SVM), and linear discriminant analysis (LDA) for all nine subjects.

Classifier	Sub1	Sub2	Sub3	Sub4	Sub5	Sub6	Sub7	Sub8	Sub9
SVM	78.90%	76.70%	66.70%	71.50%	72.00%	72.80%	73.50%	75.70%	77.40%
k-NN	77.01%	74.40%	68.30%	70.60%	73.50%	74.10%	72.02%	73.50%	84.80%
LDA	64.30%	66.30%	63.70%	66.30%	66.70%	65.20%	65.60%	67%	67.60%

**Table 2 sensors-22-01932-t002:** Accuracy, precision, and recall of deep-learning (DL) algorithm, convolutional neural network (CNN) for all nine subjects.

CNN	Sub1	Sub2	Sub3	Sub4	Sub5	Sub6	Sub7	Sub8	Sub9
Accuracy	95.47%	88.10%	85.71%	87.72%	95.29%	85.63%	85.70%	87.37%	85.52%
Precision	90.78%	86.65%	88.28%	82.94%	93.72%	86.18%	79.32%	85.23%	83.79%
Recall	87.88%	80.74%	84.37%	85.63%	90.49%	82.87%	82.60%	88.06%	81.63%

**Table 3 sensors-22-01932-t003:** Accuracy, precision and recall of deep learning (DL) algorithm, long short-term memory (LSTM) for all nine subjects.

LSTM	Sub1	Sub2	Sub3	Sub4	Sub5	Sub6	Sub7	Sub8	Sub9
Accuracy	83.81%	82.84%	82.72%	81.83%	95.35%	83.04%	81.72%	82.00%	84.81%
Precision	78.24%	83.36%	80.92%	80.83%	90.76%	85.49%	80.29%	81.43%	82.45%
Recall	80.04%	82.32%	81.75%	81.25%	93.45%	84.35%	81.82%	83.63%	79.83%

**Table 4 sensors-22-01932-t004:** Accuracy, precision and recall of deep learning (DL) algorithm, bidirectional LSTM (Bi-LSTM) for all nine subjects.

Bi-LSTM	Sub1	Sub2	Sub3	Sub4	Sub5	Sub6	Sub7	Sub8	Sub9
Accuracy	95.54%	83.55%	81.81%	82.42%	93.28%	81.67%	81.85%	82.62%	83.42%
Precision	90.74%	80.23%	82.45%	81.72%	95.56%	80.48%	84.90%	80.53%	85.37%
Recall	92.38%	82.08%	80.76%	83.62%	91.49%	82.43%	83.73%	84.29%	80.97%

**Table 5 sensors-22-01932-t005:** Statistical significance between CNN and all other five classifiers accuracies.

Classifiers	Bonferroni Correction Applied (*p* < 0.008)
CNN vs. SVM	1.42 × 10^−5^
CNN vs. k-NN	8.63 × 10^−5^
CNN vs. LDA	4.01 × 10^−12^
CNN vs. LSTM	5.35 × 10^−9^
CNN vs. Bi-LSTM	2.19 × 10^−8^

## Data Availability

The dataset presented in this study is available upon the request from the corresponding author.

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
