# Peer review of "Analyzing Classification Performance of fNIRS-BCI for Gait Rehabilitation Using Deep Neural Networks"

_sensors, 2022, doi:10.3390/s22051932_

Round 1

Reviewer 1 Report

In this study, the authors assessed the performance of different machine learning and deep learning models in gait rehabilitation using fNIRS-BCI. A promising performance was achieved, however, there are still some major concerns as follows:

1. English language should be improved. The current version contains grammatical errors and typos.

2. An important concern is the small sample size used in this study. This small amount will let the study contain more biases. Thus, it is suggested to increase the sample size.

3. Also, the authors should have some external validation data.

4. Cross-validation should be conducted to see consistent results.

5. How did the authors perform the hyperparameter tuning of all models?

6. The authors should compare the predictive performance to previous studies on the same problem/data.

7. Only accuracy is not enough to assess the model's performance. At least the authors should provide more measurement metrics.

8. More discussions should be added.

9. Deep learning or CNN has been used in previous biomedical studies i.e., PMID: 31380767. Thus, the authors are suggested to refer to more works in this description to attract a broader readership.

10. Source codes should be released for replicating the methods.

11. Quality of figures should be improved significantly.

12. In the title, the use of "deep learning networks" is not clear. Normally, people use "deep learning" or "deep neural networks".

Author Response

Response to Reviewers’ Comments

Manuscript Title: Analyzing Classification Performance of fNIRS-BCI for Gait

Rehabilitation using Deep Learning Networks

Authors: Huma Hamid, Noman Naseer *, Hammad Nazeer, Muhammad Jawad Khan,

Rayyan Khan, Umar Shahbaz Khan

Authors are really grateful to the reviewer for the thorough review of the manuscript and providing valuable comments to improve the quality of this manuscript. The manuscript has been thoroughly revised upon the comments and suggestions of the reviewer. The authors’ point-by-point answers to the comments are provided below.

Comment 1: English language should be improved. The current version contains grammatical errors and typos

Response: With reference to your kind comment, the manuscript has been thoroughly revised to improve the linguistic quality, including grammatical errors and typos, academic writing style and sentence structure.

Comment 2: An important concern is the small sample size used in this study. This small amount will let the study contain more biases. Thus, it is suggested to increase the sample size.

Response: The data was acquired in Pusan National University, Busan, Republic of Korea and the study was conducted in Air University, Islamabad, Pakistan. Authors are thankful to Professor Keum-Shik Hong for his support and cooperation. Moreover, the results are also validated on open access database containing fNIRS signals of 20 subjects, details are discussed in heading 5.3 Validation.

Comment 3: Also, the authors should have some external validation data.

Response: Following the reviewer’s comment, the analysis has been performed on  an open access database containing fNIRS brain signals of 20 subjects for the dominant foot tapping vs rest. The database contains 25 trials for each subject. The results validated the high performance of proposed method over conventional method, details are discussed in heading 5.3 Validation. Thank you for the valuable suggestion.

Comment 4: Cross-validation should be conducted to see consistent results.

Response: 10-fold Cross-validation was used for ML classifiers, while for deep learning the twenty percent of validation set were employed for each subject using automatic verification dataset and the results were consistent. This detail is also mentioned in section 5.2.

Comment 5: How did the authors perform the hyperparameter tuning of all models?

Response: Hyperparameter tuning was done through extensive trial and error method, and the selected optimal value of hyper-parameters resulted in best performance of model.

Comment 6: The authors should compare the predictive performance to previous studies on the same problem/data.

Response: Thank you! This is a very good suggestion and it will highlight the better performance of proposed methodology. The predictive performance of the previous studies on the same problem/data has been discussed in the discussions, section 6.

Comment 7: Only accuracy is not enough to assess the model's performance. At least the authors should provide more measurement metrics.

Response: Accuracy is one of the measures to evaluate performance of BCI system. However, as per reviewer’s recommendation precision and recall are also aggregated for the assessment of the model's performance in updated Table 2,3 and 4. Analysis of recall and precision also supports the conclusion and CNN performed better as compared to other DL algorithms.

Comment 8: More discussions should be added.

Response: The manuscript has been revised and more discussion has been added. Following the reviewer’s comment, separate discussion section has been added. Thankyou

Comment 9: Deep learning or CNN has been used in previous biomedical studies i.e., PMID: 31380767. Thus, the authors are suggested to refer to more works in this description to attract a broader readership.

Response: Thank you for the kind comment, mentioned and  additional references of deep learning are referred for the broader readership.

Comment 10: Source codes should be released for replicating the methods.

Response: The source codes has been uploaded to https://github.com/HumaHamid92/humahh; are available on the mentioned link for review and further use. Than you

Comment 11: Quality of figures should be improved significantly.

Response: With reference to the reviewers’ comment, visibility and quality of the figures has been improved.

Comment 12: In the title, the use of "deep learning networks" is not clear. Normally, people use "deep learning" or "deep neural networks".

Response: Thank you for the kind comment, the title has been revised from "deep learning networks" to “deep neural networks”.

Reviewer 2 Report

Comments

  1. The title mentioned that the proposed method is for “Gait Rehabilitation”, however, I did not find any references from a review paper that related to rehabilitation. The following papers can be considered:
  • A review of the progression and future implications of brain-computer interface therapies for restoration of distal upper extremity motor function after stroke.
  • Review on motor imagery based BCI systems for upper limb post-stroke neurorehabilitation: From designing to application.
  • Overview: Types of Lower Limb Exoskeletons
  1. Please remove a claim about “a novel” in the Abstract because the idea is not actually novel.
  2. In this paper, the fNIRS-BCI is explained clearly, but it will be helpful for the readers if you provide a brief description of the different brain signal acquisition techniques.
  3. It is required to check lines number 86 and 104, while the MLP classifier is defined as a conventional ML algorithm as well as DL algorithms.
  4. Line number 133, a reference is required.
  5. Visibility is not clear in Figure 4.
  6. For traditional ML algorithms, 5 statistical values are calculated, but why 3 features are used as features? (check line number 169-173)
  7. Please include some Figures of the experimental setup data acquisition.
  8. Section 2.1 described the 9 healthy right-handed male subjects were selected. I suggest adding more subjects to the experiment and including unhealthy subjects. This is to prove that the proposed method was effective.
  9. Please remove the grey background of Figure 5 and delete the title of Figure 5.
  10. Line number 188-191, sentences are not complete; please clear your statement.
  11. Line number 192-193, what does mean by notation T and n in equation 5.
  12. Notation representations are not consistent in this paper; for instance, in line number 286, e is represented with plain text, while k is represented with italic form.
  13. In Table 1, CNN vs. SVN is presented two times; why? Please also do not separate Table 1 into 2 pages.
  14. Tables 2 and 3 also need to be presented on 1 page instead of separated into 2 pages.

Author Response

Response to Reviewers’ Comments

Manuscript Title: Analyzing Classification Performance of fNIRS-BCI for Gait

Rehabilitation using Deep Learning Networks

Authors: Huma Hamid, Noman Naseer *, Hammad Nazeer, Muhammad Jawad Khan,

Rayyan Khan, Umar Shahbaz Khan

Authors are really grateful to the reviewer for the thorough review of the manuscript and providing valuable comments to improve the quality of this manuscript. The manuscript has been thoroughly revised upon the comments and suggestions of the reviewer. The authors’ point-by-point answers to the comments are provided below.

Comment 1: The title mentioned that the proposed method is for “Gait Rehabilitation”, however, I did not find any references from a review paper that related to rehabilitation. The following papers can be considered:

A review of the progression and future implications of brain-computer interface therapies for restoration of distal upper extremity motor function after stroke.

Review on motor imagery based BCI systems for upper limb post-stroke neurorehabilitation: From designing to application.

Overview: Types of Lower Limb Exoskeletons

Response: Authors are thankful to the reviewer for the thorough review of the manuscript. Thank you for the kind comment, mentioned and  additional references have been added to the manuscript.

Comment 2: Please remove a claim about “a novel” in the Abstract because the idea is not actually novel.

Response: Following the reviewer’s comment abstract has been revised. Thank you

Comment 3: In this paper, the fNIRS-BCI is explained clearly, but it will be helpful for the readers if you provide a brief description of the different brain signal acquisition techniques.

Response: For reader’s better understanding other non-invasive modalities are also included in the introduction section. Thank you for the valuable comment.

Comment 4: It is required to check lines number 86 and 104, while the MLP classifier is defined as a conventional ML algorithm as well as DL algorithms

Response: With reference to the reviewers’ comment, line number 86 and 104 has been checked and revised in the manuscript.

Comment 5: Line number 133, a reference is required.

Response: Thank you for the kind comment, reference in line following mentioned line has been added in the manuscript.

Comment 6: Visibility is not clear in Figure 4.

Response: With reference the reviewers’ comment, Figure 4 visibility and brightness has been improved.

Comment 7: For traditional ML algorithms, 5 statistical values are calculated, but why 3 features are used as features? (check line number 169-173)

Response: According to the literature, mean, peak and slope feature-combination were found to be an optimal feature-combination yielding maximum classification accuracy for fNIRS-BCI [54]. Following the literature this feature combination was used for the classification.

Comment 8: Please include some Figures of the experimental setup data acquisition.

Response: Since this data was acquired in Pusan National University, Busan, Republic of Korea and the study was conducted in Air University, Islamabad, Pakistan. Authors are thankful to Professor Keum-Shik Hong for his support and cooperation.

Comment 9: Section 2.1 described the 9 healthy right-handed male subjects were selected. I suggest adding more subjects to the experiment and including unhealthy subjects. This is to prove that the proposed method was effective.

Response: The data was acquired in Pusan National University, Busan, Republic of Korea and the study was conducted in Air University, Islamabad, Pakistan. Authors are thankful to Professor Keum-Shik Hong for his support and cooperation. Moreover, the results are also validated on open access database containing fNIRS signals of 20 subjects. (see section 5.3). Thank you for the valuable comment.

Comment 10: Please remove the grey background of Figure 5 and delete the title of Figure 5.

Response: Following the reviewers’ comment the grey background and the title of Figure 5 has been removed.

Comment 11: Line number 188-191, sentences are not complete; please clear your statement.

Response: In the following mentioned lines were checked and revised in the manuscript.

Comment 12: Line number 192-193, what does mean by notation T and n in equation 5.

Response: In the following mentioned lines the notation T means transpose where n is the sample size, this is also revised in the manuscript.

Comment 13: Notation representations are not consistent in this paper; for instance, in line number 286, e is represented with plain text, while k is represented with italic form.

Response: Following the reviewer’s comment, the manuscript has been thoroughly revised to remove the inconsistency in the text. Thank for the correction.

Comment 14: In Table 1, CNN vs. SVM is presented two times; why? Please also do not separate Table 1 into 2 pages.

Response: Following the reviewers’ comment, Table 1 has been revised since, there was a mistake of the repetition in values. Thank you for the correction

Comment 15: Tables 2 and 3 also need to be presented on 1 page instead of separated into 2 pages.

Response: Following the reviewers’ comment, Tables 2 and 3 has been revised. Thank you for the correction

Round 2

Reviewer 1 Report

My previous comments have been addressed.

Reviewer 2 Report

Dear Authors,

Thank you for providing the revision documents. I have checked the revised paper and I have no further comments.

- Reviewer -